# Advances in Fabricating the Electrospun Biopolymer-Based Biomaterials

**DOI:** 10.3390/jfb12020026

**Published:** 2021-04-16

**Authors:** Sebastian Wilk, Aleksandra Benko

**Affiliations:** Faculty of Materials Science and Ceramics, AGH University of Science and Technology, A. Mickiewicz 30 Avenue, 30-059 Krakow, Poland; sewilk@agh.edu.pl

**Keywords:** electrospinning, biopolymers, tissue engineering, crosslinking strategies, peptides

## Abstract

Biopolymers formed into a fibrous morphology through electrospinning are of increasing interest in the field of biomedicine due to their intrinsic biocompatibility and biodegradability and their ability to be biomimetic to various fibrous structures present in animal tissues. However, their mechanical properties are often unsatisfactory and their processing may be troublesome. Thus, extensive research interest is focused on improving these qualities. This review article presents the selection of the recent advances in techniques aimed to improve the electrospinnability of various biopolymers (polysaccharides, polynucleotides, peptides, and phospholipids). The electrospinning of single materials, and the variety of co-polymers, with and without additives, is covered. Additionally, various crosslinking strategies are presented. Examples of cytocompatibility, biocompatibility, and antimicrobial properties are analyzed. Special attention is given to whey protein isolate as an example of a novel, promising, green material with good potential in the field of biomedicine. This review ends with a brief summary and outlook for the biomedical applicability of electrospinnable biopolymers.

## 1. Introduction

Biopolymers are a group of materials derived from natural and renewable sources that attract constantly increasing attention from various fields in the ongoing search for an “eco-friendly” alternative that can alleviate the dependence on extraction and processing of fossil fuels. The animal or plant origin of polymers not only translates into a highly ecological approach, but also provides extremely beneficial properties of the material, particularly high biocompatibility, biodegradability and, potentially, antimicrobial properties. Biopolymers are usually inexpensive, non-toxic, nutrient-rich, and edible. For these reasons, the majority of current applications are focused around the food industry [1,2]. Furthermore, there has been an increase in the use of biopolymers in other fields of industry and science, including medicine where biopolymers are recognized as promising materials in the production of advanced scaffolds and substrates for cell cultures. In this respect, the main advantage of the biopolymers over synthetic materials lies in their highly biomimetic nature, which can provide cells with an environment similar in morphology and chemical structure to the natural extracellular matrix (ECM). Thus, biopolymers can stimulate cellular responses similar to those observed in living tissues. Their properties and potential applications can be further enhanced and expanded by controlled functionalization and/or modification with various biochemical factors [3]. Unfortunately, biopolymers also have some disadvantages, the most important being that natural materials show significant differences in morphology and chemical structures, even when they originate from the same production batch. There is also a hard-to-mitigate risk of various pathogen transfer, arising from improper cleaning after harvest from an animal or a plant source. To reduce this risk, complicated, multi-step procedures that are both costly and time-consuming [2,3] are implemented. Nevertheless, due to numerous advantages and desirable physicochemical properties, the number of studies involving the biopolymers is constantly increasing. Among them, the fabrication of nanofibrous scaffolds via electrospinning (ES) is particularly interesting. This is due to the numerous advantages of this nanoscaled morphology (e.g., high surface to volume ratio, biomimetism, ability to interact with the cells at a molecular level) combined with the high versatility, cost-efficiency, and simplicity of the ES technique.

Since electrospinning can be used to produce fibrous materials of controllable properties, this process is often the method of choice for fabricating novel and biomimetic cell culture scaffolds. However, it also poses many challenges, some of which are very difficult to mitigate—low spinnability potential, issues with maintaining the biopolymer’s native conformation, and proper crosslinking to obtain a water-insoluble material—in an effort to optimize the process.

This paper aims to gather recent advances and achievements in the field of the ES of biopolymers, starting with a justification for the need to obtain fibrous scaffolds, and then providing a basic description of the process, followed by an identification of the most important challenges in the ES of biopolymers with various strategies to overcome them. Next, various groups of electrospinnable biopolymers are discussed, with selected examples that are important from a biomedical point of view. Finally, whey protein isolate (WPI), an entirely green and biocompatible biopolymer, is discussed as a likely candidate to progress into biomedicine.

Even though the ES of biopolymers is attracting constantly increasing attention, review articles focusing on this matter are rather scarce and hardly ever discuss the strategies employed to improve the electrospinnability for their further use in biomedical applications. Furthermore, the need to maintain the biopolymers’ native conformation is often not sufficiently stressed. Fortunately, few examples of important literature on the subject can be found. Possibly two of the first reviews on the matter are a 2006 book chapter by Gisela Buschle-Diller et al. [4] and a 2008 study by Jessica D. Schiffman and Caroline L. Schauer [1]. While providing good background information and some basic guidance for making the biopolymers electrospinnable, these older summaries lack information about the advances in the field and do not identify the necessity of maintaining the biopolymers’ native conformations and use of the benign solvents/crosslinking agents in the process. In a more recent study from 2018 by Soares et al. [5], the authors have critically evaluated the available literature, identifying the most promising fields for applications of electrospun biopolymers. Covering a very broad spectrum of subjects, the article did not analyze the strategies employed to make mono-biopolymers electrospinnable, nor did it mention the crosslinking methods. Additionally, various classes of biopolymers with a focus on their biological performance were not evaluated. In 2019, two articles reviewed the use of various electrospun biopolymers in the field of food packaging [6] and encapsulation of food bioactive ingredients to improve their stability during oral supplementation [7]. Both of these studies provided excellent guidance on strategies employed to electrospin the biopolymers, including the use of a variety of solvents and protein denaturation to improve spinnability. However, the subject was not extensively evaluated; fiber stability and morphology were not analyzed, and the biomedical applications were not discussed. Finally, two very recent reviews have described the biomedical applicability of electrospun biopolymers. The specific fields of application were wound healing [8] and tissue engineering [9]. Both of the articles identified the need to produce scaffolds biomimetic to the tissues’ natural ECM and analyzed the importance of ES parameters. In the research presented by Bombin et al. [8], an extensive evaluation of electrospinnable biopolymers, along with their biological performance, is presented, focusing on the in vivo behavior. Additionally, the authors stressed the need for process optimization. Unfortunately, the study does not identify the need to maintain the biopolymers’ native conformation and the desire to reduce the use of organic solvents and/or potentially toxic crosslinkers as part of “green” process improvement. Meanwhile, a study by Rahmati et al. [9] provided an extensive evaluation of solution-related ES parameters, and discussed the application of electrospun materials in the regeneration of various tissue types, and provided valid guidance for surface modification strategies; however, most of the materials presented in the study were blends with synthetic materials, with or without biopolymer addition. Finally, an article by Jain et al. [10] summarized the properties of electrospinnable biopolymers in terms of their applicability for drug delivery systems. Different processes affecting parameters were broadly analyzed and strategies for drug loading were listed, along with examples of electrospinnable polymers. The largest focus was placed on polysaccharides and their biological performance, being specifically evaluated in terms of drug delivery efficiency.

To summarize, even though there are multiple sources describing the ES of biopolymers, to the best of our knowledge, no single all-encompassing review exists that provides a thorough guidance on how to obtain biopolymer-based fibrous scaffolds, designed for use in various fields of medicine. The above-mentioned reviews fail to identify the importance of reducing the use of possibly harmful and environmentally damaging byproducts such as solvents, crosslinkers and additives in the search for materials with desired morphologies. More importantly, the materials’ stability is hardly ever analyzed. This review was written with the objective of filling these gaps, providing extensive guidance on the current strategies employed to create electrospinnable materials, including crosslinking, both in terms of maintaining the biopolymers’ native conformation, which can guarantee superior biomedical performance, and in achieving desirable stability. We believe that this review can serve as a stepping stone for scientists focusing on the subject and leading to a more systematic approach to further development. The focus of the presented review is centered around peptides, their being the main constituents of the tissues’ fibrous ECM, and representing the most interesting and the most studied group of materials.

## 2. The Advantages of Forming Biopolymers into Fibrous Morphology

The key challenge in tissue engineering and cell culture is to design and fabricate the tissue-specific scaffold that would fulfill the need for chemical, biological, and morphological biomimicry, combined with biocompatibility, biodegradability, and optimal mechanical properties. Among the variety of approaches, fabrication of scaffolds with a high similarity to the tissue’s natural ECM seems to be the most promising. In the living organisms, ECM creates the natural environment for cells and tissues, thus affecting and controlling the cellular response and neo-tissue genesis. The main building blocks of ECM are proteins such as collagen, elastin, fibronectin, or laminin, all being fibrous in morphology. Therefore, using an appropriate fibrous structure in the manufacturing of the scaffolds for cell deposition creates a potential to mimic the natural ECM morphological functionality, guiding the cell adhesion, proliferation, differentiation, and cell orientation. In addition, the use of biopolymers introduces good biocompatibility, biodegradability, and biochemical mimicry. The carefully designed biopolymers can contain specific motifs, such as RGD (arginine, glycine, and aspartate tripeptide), leading to further promotion of the cellular adhesion. Apart from the cellular-control features, the fibers, in particular those of small diameters (up to few hundreds nanometers), are characterized by large surface-to-mass and surface-to-volume ratios, high porosity, and superior mechanical performance, when compared to the alternate morphologies. The high surface-to-volume ratio introduces a larger area for scaffold–cell–tissue interaction. The increase in porosity improves the desired tissue ingrowth. Depending on the manufacturing process, the individual fibers can be obtained with diameters ranging from the nanometric to the millimetric scale, enabling the reconstruction of hierarchical tissues at various levels. In addition, the high spectrum of the possibilities in the shaping of the fibrous scaffolds, control of the fiber orientation in volume, and ease of the structure modification, greatly increase the range of the potential biomedical applications [2,11,12].

## 3. Electrospinning—A Versatile Technique for Fibrous Material Manufacturing

Electrospinning (ES) is one of the most common and versatile methods employed in the manufacturing of fibrous materials, under laboratory and industrial-scale production. The ES process allows for precise control of the physicochemical parameters, such as morphology and diameter of the fibers, their orientation, and distribution in volume, as well as porosity and thickness of the final product. In addition, it is relatively inexpensive and does not require the use of specialized equipment. The most basic ES device consists of a syringe filled with the polymer solution (usually connected to the syringe pump), a metal needle connected to the high voltage supply, and a collector, which is either grounded or connected to a second voltage source of an opposite pole (Figure 1). During the needle-based electrospinning process, the polymer solution is being pushed by the pump from the syringe at a constant speed. At the tip of the needle, the solution is being exposed to an electric field, which is created between the needle and the collector. In an alternate approach—the needleless ES—the polymer solution is placed in the bath and a rotating mandril, which is connected to the high voltage source, is immersed into it. The mandril collects a thin layer of the polymer solution that is then exposed to an electric field upon facing the grounded collector electrode. Either at the tip of the needle or from the thin layer of the polymer solution at the rotating mandril, the electrostatic forces of the field overcome the surface tension of the solution, forcing it to create the Taylor cone (or multiple Taylor cones). From these cones, the charged polymer jet is ejected and elongated on its route to the collector. During this process, a solvent evaporates from the solution, thinning the continuous jet of pure polymer, and is deposited on the collector in a fibrous form [13,14].

The electrospinning technique allows the user to create ultrafine fibers of submicron diameters, in a continuous matter. It also enables the generation of the advanced, three-dimensional scaffolds, with the shape being controlled by the selection of an appropriate collector type (more details can be found in a study by W. E. Teo and S. Ramakrishna [15]). Unfortunately, the biggest drawback of the ES technique is that for each polymer-solvent system, a scrupulous optimization of the manufacturing parameters must be performed to properly control the process and the resultant final product properties. Since the polymers and their solutions differ from each other in viscosity, electrical conductivity, and other important properties, their behavior during ES varies, even when the same conditions are employed. The unoptimized process can result in undesired material defects, such as beads and droplets, a lack of the product reproducibility, or, in extreme cases, a complete lack of solution spinnability.

The controllable process parameters can be divided into three categories: solution-related, device-related, and ambient. The most important examples from each category are presented in Figure 2 (based on [16,17]). An excellent evaluation of the ES-affecting parameters can be found in a study by Bombin et al. [8] and in a recent review by Xue et al. [18]

The solution-related parameters are the most challenging to optimize as they depend on the polymer and solvent type, presence of the additives, molecular arrangement and entanglement in the solution, and, in some cases, even on the employed handling procedures. It has been recently identified that at the specific solution viscosity, related to the solution concentration, the molecules of sufficiently high molecular weights (for example, 13,000–23,000 g/mol in case of PVA [8]), can form the so-called semi-dilute II, entangled regime [9]. In such a state, the solution exhibits the viscoelastic behavior and the molecules are able to form the continuous, stable fibers that are vital for production of the high-quality ES mats. This phenomenon is used to determine the entanglement concentration—a parameter that defines the minimum concentration at which a certain polymer can be electrospun [8,9,19]. In theory, above the entanglement concentration, the solution can be spun if the minimum voltage able to form the Taylor cone is applied. In practice, however, this is not as straightforward for the biopolymers as it is for the synthetic ones. This is because the former tend to be nonuniform in terms of their molecular weight.

Another issue affecting the polymer–solution interactions is the tendency of some of the materials to assemble into the larger aggregates or to form chemical interactions with different components of the solution. Chemical interactions that take place in the solution, both those that occur spontaneously and those that are intentionally introduced, have been extensively evaluated and described in a recent study by Elena Ewaldz and Blair Brettmann [20]. The polymers that tend to form such interactions do not obey typical laws established for the ES—for example, the thickness of the fibers cannot be strictly controlled, the gelling can occur or generally non-spinnable solutions can become spinnable. One example of naturally occurring self-assembling biopolymers are phospholipids [21], but this phenomenon is also observed in cyclodextrins or tannic acid [20]. It is worth noting that knowledge about self-assembly can be harvested for designing the materials to meet the specific needs. In 2013, Tayi et al. performed a successful ES from very low concentration (4%) water solutions of peptide amphiphiles (molecules possessing both hydrophilic and lypophilic properties), claiming its applicability in medicine [22]. In a newer study, Yao et al. [23] generated scaffolds for the neural tissue engineering. Benefiting from fibrous and aligned morphology, and the soft gel mechanical properties obtained by self-assembly, the scaffolds were capable of inducing the stem cell differentiation along the neural lineage. The substrate of choice was fibrin hydrogel, and it was hypothesized that the ES induces the internal alignment followed by the chemical crosslinking. The material was easy to handle and provided an excellent scaffold for inducing cellular adhesion, alignment, neural outgrowth, and differentiation. In in vivo studies performed on a rat T9 dorsal hemisection spinal cord injury model, the scaffold promoted endogenous cell invasion and proliferation, forming aligned tissue cables. In a follow-up study [24], the same procedure was employed to create a scaffold inside a chitosan tube for better stability, and the resulting scaffold was implanted into the rats, which had a 7 mm fragment of the sciatic nerve surgically removed. Evaluated material promoted the complete nerve regeneration, restoring the electrophysiological and the motor functions to a similar extent to that of the autograft. Overall, the material proved to be a promising candidate for neural tissue engineering, benefiting from the combination of self-assembly and ES.

Another important aspect in the biopolymers’ ES is the control of conformation. A large share of biopolymers, in particular proteins, have a native globular structure. This type of structure limits the molecule’s ability to form the intramolecular interactions. Hence, the material is not able to create an entangled regime needed to achieve the spinnability. For these types of materials, structural denaturation can be necessary, so that the material can be unfolded to interact with other molecules present in the solution [6,7].

## 4. The Key Challenges in Electrospinning of the Biopolymers

The processing and electrospinning of the biopolymers can be challenging, due to their native properties. First of all, as mentioned before, polymers of the natural origin may significantly differ in their properties, such as molecular weight, purity, electric charge distribution, and crystallinity from batch to batch. Unfortunately, those are at the same time the main parameters impacting the electrospinning process and, as a consequence, each batch of the biopolymer solution for ES may require an independent optimization, as had been described in the review from 2008 [1].

The selection of the proper solvent is typically the first step on the long route towards the ES optimization. Typically, organic solvents are chosen as first-line candidates due to their large dipole moment, low viscosity, high evaporation rate, and very high solubility of most polymers, ensuring the versatility in the manufacturing of the reproducible, high quality fibers of uniform diameter [25,26]. In the case of biopolymers, however, especially for the medical applications, the use of the organic solvents is undesired. While some of the biopolymers are completely insoluble in organic solvents (e.g., chitosan [27], or cellulose [28,29]), others, like collagen, can be irreversibly damaged as their structure can be permanently denatured [30]. In addition, organic solvents tend to be toxic to living organisms and the ES products may contain some solvent residues. This poses a serious health risk, reducing the chance of material to progress into the clinical trials. Therefore, when possible, the use of organic solvents in the field of food and tissue engineering should be avoided. Ionic liquids [28], aqueous solutions of salts (inorganic electrolytes) [29,31], organic electrolytes [29], and various acidic solutions [27,32] are suggested in some cases as more friendly alternatives.

The use of organic and inorganic electrolytes can play an important role in mitigating the risk of strong hydrogen bond formation, which results in the high viscosity of the solution, leading to difficulties in formation of stable, polymer jets in the electric field [31]. Alternatively, biopolymers can be chemically modified to introduce the repulsive interactions between the chains, thus reducing the solution’s viscosity [32].

Lastly, the ES of biopolymers can be impacted by their sub-ideal mechanical properties, disrupting the continuous fiber formation, and making the final product very fragile and easily susceptible to physical damage [2]. Such defects can be overcome by either (1) introducing synthetic or natural co-polymers that are able to form additional interactions between the chains [33,34]; or (2) performing a chemical modification of the chains for successive self-cross-linking [35,36,37].

As the challenges impacting the biopolymer ES processes are widely known, there are numerous ways to decrease the risk and improve the success of biopolymer electrospinning. Examples of such approaches have been gathered in Table 1. Among them, the most widely used one is to combine the biomaterial in solution, with other, well-spinnable polymers of a synthetic or natural origin (including other biopolymers). This technique does not only enable a stable and effective electrospinning process, but it also improves the mechanical properties of the obtained composite, and, in some cases, even increases the biocompatibility of the final product. The physicochemical properties of the supporting material have to be carefully chosen to complement or eliminate the deficiencies of the natural polymer, and therefore enable the production of fibers with the desired diameter, arrangement, orientation, and morphology [38,39].

Another critical aspect in the electrospinning process of biopolymers is maintaining their conformation, which is crucial in assuring their proper bioactive performance. The solvents applied, crosslinking procedure, ES conditions, as well as usage of additives, may result in damage to the material’s native bio functionality, resulting in a much lower level of applicability than originally expected. Hence, even upon successful processing of the biomaterial into a fibrous morphology, its performance might not necessarily meet the desired specification as it has been thoroughly described in a study by Campiglio et al. [45].

## 5. The Crosslinking Strategies

In general, the as-obtained electrospun all-biopolymer matrices are mechanically weak and water-soluble. These effects are strongly undesired and yet are even more damaging at an elevated temperature in the human body. Therefore, adequate structural crosslinking is necessary to stabilize the fibers and ensure that the designed morphology and the favorable properties will be preserved. The crosslinking can be achieved by numerous methods, which can be divided into either physical or chemical categories. The first type involves the application of various external stimuli, while the second one is performed with the use of chemical substances [46]. Physical crosslinking might be achieved through thermal treatment [47,48], electron-beam irradiation, or plasma treatment [46]. While most physical techniques are considered as “green” methods, they may have a lower efficiency in the case of thick, multilayered materials. Meanwhile, chemical treatment strategies usually ensure higher efficiency, leading to significantly better properties of the final product. Glutaraldehyde (GTA) vapor seems to be one of the most popular and well-described ways to chemically crosslink various biopolymers [1,2,46]. However, it is worth noting that, despite its numerous advantages, evidence of GTA cytotoxicity had been presented [49,50], deeming it undesirable in biological applications.

Thus, other substances, e.g., cinnamaldehyde (CA) [48], aqueous salt solutions, e.g., CaCl_2_ [51], citric acid [52], and agents such as genipin (GP) or N-3-dimethylaminopropyl-N’-ethylcarbodiimide hydrochloride (EDC), are used [46,53]. A great comparison of the impact of various, widely used crosslinkers on the process of denaturation of the collagen fibers can be found in the paper by Luo et al. [50] As the authors have reported, while all evaluated crosslinkers (GP, GTA, and EDC-N-hydroxysuccinimide (NHS)) were effective in inhibiting the materials’ solubility over 2 months of incubation (while the un-crosslinked material had completely dissolved), GTA reduced the material’s cytocompatibility, while GP or EDC both were less effective in maintaining intact fibrous morphology (Figure 3) [50], leading to the conclusion that there is no universal chemical crosslinker that would combine excellent biocompatibility with an ability to maintain an intact fibrous morphology. Instead, one is forced to balance between those two qualities. Hence, there is a need to establish a new, green, and biocompatible array of crosslinkers that would not be a detriment to the fibrous morphology of the electrospun biopolymers.

UV irradiation is often considered as a chemical crosslinking method, as it is achieved through chemical photo initiators (either created in-situ or introduced from an exogenous source) that initiate the catalytic process of the formation of the covalent bonds between the polymer chains [46]. Among different exogenous photo initiators (also referred to as crosslinker), the Irgacure 2959 (1-[4-(2-hydroxyethoxy)phenyl]-2-hydroxy-2-methyl-1-propan-1-one) seems to be the most popular one [54,55]. Unfortunately, as many others active chemical compounds, it can also be considered to be an undesirable additive. The most promising area of the UV light application is the UV-induced crosslinking performed without the use of exogenous photo initiators—by modifying the polymer backbone with specific functional groups. An example of such an approach can be found in the study by Seo et al. [41], who fabricated zwitterionic 2-methacryloy-loxyethyl phosphorylcholine (MPC), containing photo-cross-linkable phenyl azide groups. The synthesized fibrous membranes were successfully crosslinked by the UV light exposure while maintaining their fibrous morphology [56]. This is an interesting approach as it reduces the risk of possibly toxic crosslinker residues which could be leaching from the scaffolds throughout their life span.

## 6. Assuring Proper Physicochemical Properties of the Electrospun Biopolymers

Proper quality control of the manufacturing process, as well as extensive analysis of the resulting electrospun biomaterials, are critical in obtaining materials of desired properties, including biological performance. In addition to the polymer solution properties [57] playing important role in the adequate control of the manufacturing process, the final product should be thoroughly investigated as well. Morphology of the fibers, their chemical composition and crystallinity level, combined with the conformation and evaluation of the presence of bioactive compounds, should always be kept in mind and properly assessed. The evaluation of the morphology can be performed by a scanning electron microscopy, while the chemical composition can be tested with infrared spectroscopy (FTIR) or mass spectroscopy. Crystallinity can be directly analyzed by the differential scanning calorimetry technique, or indirectly qualitatively observed through mechanical tests. To some extent, FTIR can also provide some information about the crystal form of the sample. One of the most challenging properties to analyze is the biopolymer’s secondary and tertiary structure (conformation) as the methods that can properly evaluate it are limited. The secondary structure is often analyzed by the circular dichroism spectroscopy, which measures the molecule’s chirality, or through FTIR, which identifies the changes in the amount and type of functional groups present in the structure [58]. Meanwhile, the tertiary structure can be evaluated by the sodium dodecyl sulfate polyacrylamide gel electrophoresis [44], X-ray diffraction or nuclear magnetic resonance [59]. Comprehensive guidance through the analytical techniques used to investigate the biopolymer fibrous materials’ properties can be found in an excellent review by Ricaurte Leidy and Quintanilla-Carvajal Maria Ximena [57]. Some approaches and guidance to properly evaluate the fibrous materials are also listed in the summary by Xue et al. [18].

## 7. Examples of the Electrospinnable Biopolymers

To this date, various types of biopolymers have been electrospun, either as homopolymers, or in the presence of various additives. Examples of the optimization studies geared towards identification of the biopolymers’ electrospinnable windows and adequate post-processing are gathered in Figure 4.

Biopolymers can be classified based on the chemical composition. There are four main classes that can be clearly distinguished: polysaccharides, polypeptides, polynucleotides, and phospholipid polymers [60]. Below, we present the examples of the applications grouped by the structural classifications described above.

### 7.1. Polysaccharides

Polysaccharides are defined as polymers with macromolecules composed of simple sugars. This group is highly demanding and is relatively difficult to electrospin due to the complex long-range electrostatic interactions [1]. In addition, when dissolved in solvents, the sugar oligiomers can create cationic or anionic solutions, depending almost completely on the type and character of the material used [1]. Surprisingly, however, unlike other biopolymers, the polysaccharides represent quite a few examples of the materials that can be easily processed without using additives or copolymers; for example, maltodextrin (MD) and dextran (DE) (both will be mentioned and discussed in a further paragraph), as well as xanthan gum, pullulan (PUL), starch, and guar gum. Some studies also suggest that both chitosan/chitin and alginate can be electrospun without additives.

#### 7.1.1. Chitosan and Chitin

Chitosan is an example of a sugar-based polymer material obtained from chitin via deacetylation. It is likely the most popular example of a polysaccharide to be used in various biomedical applications. Due to its exceptional qualities, it is also one of the most promising green biomaterials. The extremely rigid crystal structure of this polymer results in very limited solubility in high pH water and numerous organic solvents. The solubility significantly increases in aqueous solution under acidic conditions with a pH lower than 6.5. Chitosan is non-toxic, biocompatible, and biodegradable and has intrinsic bacteriostatic and bactericidal properties. It also exhibits a high mechanical strength and a high affinity to binding proteins, making it an ideal candidate for biomedical applications. Due to its high availability, low cost, and good biological and mechanical properties, chitosan has attracted significant interest in fibrous, biomedical applications. Hence, a significant amount of comprehensive review articles already exists on the subject [61,62,63].

Unfortunately, electrospinning of pure chitosan is quite difficult and challenging due to its tendency to form extremely viscous solutions at high concentrations in a semi-dilute II, entangled regime. Therefore, various additives and solvents, such as PEO, PVA, gellan gum or 1,1,1,3,3,3-hexafluoro-2-propanol (HFIP) [2] and TFA [10] are used, to enable the stable electrospinning process of nanofibrous product. Unfortunately, the toxicity of the above-described additives and solvents is quite high. To alleviate the use of the undesired process additives, de Vrieze introduced a more benign solvent—acetic acid (90%) solution at a 3% concentration to electrospin chitosan. No biocompatibility results on the material were, however, reported [64]. Another literature approach [65,66] was centered around the use of chitosan derivatives, unfortunately, yet again, the additives and non-green solvents were employed in the process.

In addition to chitosan, also chitin, an acetylated chitosan’s protoplast, is often mentioned as a biocompatible, electrospinnable biopolymer. In a 2018 article by Jung et al. [67], β-chitin was extracted from cuttlefish bone by deproteinization and demineralization using 1 N sodium hydroxide solution, followed by treatment with 1 N hydrochloric acid solution. The resulting material was subsequently dissolved in formic acid and electrospun with and without PEO. While in both cases a good quality fibrous product was obtained, PEO improved the material’s spinnability. In an in vivo rat wound healing model, the material without PEO was found to provide better results, leading to faster wound healing. While further studies are still necessary to fully assess the properties of the material, the article is a rare example of the successful ES of chitin without the use of harmful additives, and the use of environmentally friendly solvents.

#### 7.1.2. Alginate

Alginate is a polysaccharide, linear copolymer of β-1,4-D-mannuronic acid and α-1,4-L-guluronic acid, harvested from the cell walls of brown algae or synthesized by *Azotobacter* and *Pseudomonas* bacteria strains. The alginate possesses numerous valuable properties, such as biodegradability, biocompatibility, high hygroscopicity, antimicrobial properties, and high ion adsorption. Therefore, it is of high interest in numerous applications within the biomedical field [51]. Alginates are insoluble in organic solvents but, under certain conditions, they can be dissolved in water. Due to the high viscosity arising from the fact that alginates are polyelectrolytes with a rigid intramolecular and intermolecular hydrogen network, their ES tends to be difficult. The aqueous solutions can only be spun at a very narrow concentration range, even in the presence of easily spinnable polymer additives [2].

#### 7.1.3. Starch and Its Derivatives

Starch, composed of amylose and amylopectin, is derived from various plant algae and bacteria sources. It is widely utilized in various industry branches such as food and food packaging, due to its versatility, biodegradability, cost efficiency, and worldwide abundance [43,68]. Starch can be spun without copolymers from the DMSO or DMSO/aqueous solutions, if the concentrations that exceed the entanglement of starch chains is used [43]. Soluble potato starch was successfully electrospun from formic acid/water solutions (50% of starch concentration), with the addition of carvacrol as an antioxidant, leading to the formation of the drug-loaded nanofibers of homogenous morphology, exhibiting the antimicrobial and antibacterial properties against *L. monocytogenes*, *S. Typhimurium*, *E. coli*, and *S. aureus* strains. As expected, the antioxidant properties of the material increased with the concentration of added carvacrol. The evaluation of the synthesized material indicated the potential for future use in both food and biomedical applications, such as tissue engineering and cell culture [68]. The thorough review of the starch electrospinning process was presented by Liu et al. [69] The experimental details were included, along with potential biomedical applications. In another summary by Hemamalini and Dev [70], various strategies were utilized to synthesize fibrous biomaterials out of starch; modified starch and various starch–co-polymer blends were extensively evaluated.

The next polysaccharide—pullulan (PUL)—is a linear glucan, synthesized in nature by *Aureobasidium pullulans* yeast-like fungi from starch. It is non-toxic, highly soluble in water, stable regardless of pH and temperature changes, and exhibits excellent fibers and film-forming capabilities. Its desired properties originate from the regular alterations of α-(1→4) and α-(1→6) glycosidic bonds. PUL was electrospun by Kong et al. [71] from an aqueous DMSO solution and from aqueous solutions containing salts. Li et al. [31] described the relationship between the concentrations of sodium chloride and sodium citrate in solution and hydrogen bonding between the polysaccharides and water molecules, as well as the inter- and intra-macromolecular chains. They demonstrated that small concentrations of sodium salts improve the quality of the fibrous final product, by preventing bead formation on its surface.

Cyclodextrins (CDs) are another example of electrospinnable polysaccharides. CDs are naturally occurring, water-soluble, and non-toxic cyclic oligosaccharides derived from starch in the process of enzymatic conversion. They are characterized by a toroid-shaped molecular structure, which can form non-covalent host–guest inclusion complexes (IC) with other molecules, resulting in changes to their physical and chemical properties: reduction in the volatility, improvement in the mechanical stability and/or solubility. Some examples of commonly used ICs are drugs and antimicrobial systems [72,73,74]. Due to their ability to self-assemble [72], CDs or CD-IC complexes can be electrospun from aqueous solutions, without the presence of any additive [74]. Celebioglou et al. [73] employed this property to synthesize electrospun nanofibers from a highly concentrated (160% CD, w/w) aqueous triclosan/CD-IC suspension, using two different chemically modified CD types: hydroxypropyl-beta-cyclodextrin (HPβCD) and hydroxypropyl-gamma-cyclodextrin (HPγCD). Non-volatile and hydrophobic triclosan has been incorporated to ensure the bactericidal properties and form the host–guest IC. Antimicrobial tests on *E. coli* and *S. aureus* bacteria strains were performed, proving the desired activity from both triclosan/HPβCD-IC and triclosan/HPγCD-IC fibrous mats, as compared to pure triclosan powder. Hence, the synthesized mats were regarded as interesting candidates for wound dressings and cell culture scaffolds applications. Aytac et al. [74] synthesized CD/linalool-IC nanofibers from an aqueous solution. Linalool i.e., 3,7-dimethyl-1,6-octadien-3-ol is an acyclic alcohol, known for its antimicrobial, anti-inflammatory, and anesthetic properties. The generated nanofibers had high antibacterial activity against both Gram-negative (*E. coli*) and Gram-positive (*S. aureus*) bacteria. The CD/linalool-IC complexes were dissolving in water after 2 s, indicating no-desired poor stability for initially targeted application. Therefore, synthesized materials were proposed as fast-dissolving systems for use in pharmaceuticals or in food engineering.

To summarize, even though promising results were presented, describing the ES of cyclodextrins, a lack of evidence for their prolonged stability in various conditions indicates further studies focused on enhancing this quality are necessary.

#### 7.1.4. Xanthan and Guar Gums

Xanthan gum is an extracellular heteropolysaccharide synthesized by the *Xanthomonas campestris* bacteria strain. This polymer contains glucose, mannose, and glucuronic acid in a 2:2:1 molar ratio. Xanthan gum had been electrospun with formic acid as a solvent by Shekarforoush et al. [32]. Unlike a typical “weak gel-like” substance usually formed with other solvents, the formic acid solution led to the stably spinnable material. This unusual behavior was a consequence of the esterification reaction between the formic acid and pyruvic acid groups of xanthan gum. When processed, the use of the solution led to the formation of a nonwoven mat of sub-micron fibers [32].

Guar gum is an inexpensive, abundant, and biodegradable polysaccharide derived from natural sources, and is regarded as safe in food and drug applications. Pirzada et al. electrospun an aqueous solution of high molecular weight (M_W_) native guar with its partially hydrolyzed low M_W_ analogs. Such a combination resulted in a stable electrospinning process and the generation of a fibrous mat from substances that could not be spun otherwise [41].

#### 7.1.5. Kefiran

While kefiran has been known since the 1960s, its progression into biomaterial science is fairly new. This branched polysaccharide is produced by kefir grain microorganisms. It was first electrospun in 2014 by Esnaashari et al. [75] from a simple, water-based solution, without additives. Since then, kefiran has been combined with various additives and other biopolymers, resulting in multifunctional nanofibrous scaffolds that can be utilized in fields of medicine and/or food industry. The probiotic properties have been reported to enhance the growth of favorable microflora, while, at the same time, inhibit the cancer cells and bacteria. Even though many more studies are needed to reveal its true potential and applicability, kefiran is expected to create a new, exciting class of biomaterials with superior properties [76].

#### 7.1.6. Cellulose

Cellulose has been identified as one of the first biopolymers to be synthesized into the desired morphology. This polysaccharide, composed of glucose monomers, is abundant in plants, microorganisms and marine flora [6], and can be found in large quantities (up to 90%) in various agroindustry waste products, making it an extremely interesting, environmental friendly material [77]. Pure cellulose is insoluble in water and in most organic solvents, and thus its ES is challenging but achievable under appropriate conditions. In 2005, Piotr Kulpinski [78] dissolved α-cellulose in 50% water solution of *N*-methylmorpholine-*N*-oxide (skin, respiratory and eye irritant). A stable ES process was carried out using a 2% cellulose concentration, at 90 °C, by applying a 9.5 kV voltage. As the process can only be conducted at very high temperatures (due to low solvent evaporation rate), in recent years, various alternative strategies have been developed. These often include electrospinning from ionic liquids, or use cellulose derivatives [77,79]. With its excellent biocompatibility and good mechanical properties, unmatched by other biopolymers, cellulose fibers have been identified as promising new reinforcing additives to be used in materials aimed towards the reconstruction or regeneration of various tissues, with a strong emphasis on cardiovascular applications. Potential alternate fields of applications also include drug delivery systems and wound dressings. Cellulose appears to be one of the limited biopolymers that are close to implementation into clinical applications, with long-term in vivo studies in rats revealing its superior performance as artificial valves [80].

In a recent study, Santos et al. [81] had synthesized electrospun cellulose acetate fibers, modified in volume with an annatto extract. The prepared material was evaluated as a wound dressing, able to induce the antibacterial and anti-inflammatory reactions. Even though the fibers were electrospun from the mixture of organic solvents—acetone and DMF—the process did not reduce the bioactivity of the annatto extract, yielding superior results in in vivo studies performed on rats. As suggested by the authors, immunomodulatory effects were observed through an inhibition of the polymorphonuclear cell recruitment at a wound site. Certainly, further studies are needed to evaluate the material’s long-term biocompatibility and its potential ability to enhance the tissue regeneration at the wound site, as well as to establish potential antibacterial effects.

While many studies on the biomedical applications of cellulose fibers and their derivatives exist, an unusual approach was investigated by Zhao et al. [82], who proposed a cell capture system, designed to target metastatic cancer cells that had escaped the primary tumor site and entered the blood stream system to create secondary tumors. In their application, cellulose acetate mats were prepared by electrospinning and subsequent hydrolysis, followed by covalent bonding with dendrimers. The dendrimers were modified with a cancer cell targeting ligand—folic acid. With the high surface-to-volume ratio, cancer cell selective material, with up to 100% cell capture, was efficiently obtained. While the proposed solution certainly is interesting, further studies should be performed to evaluate in vivo applicability, as aside from the 100% target cell specificity, up to 60% of non-specific binding was also observed, indicating the need for further consideration.

### 7.2. Polynucleotides

Desoxyribonucleic acid (DNA) is responsible for the storage of all genetic information about the structure and function of living organisms. This biopolymer consists of nucleotides that are built from the sugar, at least one phosphate residue, and one of the nitrogen bases. The first successful electrospinning of DNA was performed in 1997 from the water–ethanol solvent system. Pure DNA cannot be spun from pure water, and therefore it is usually combined with other polymers such as PEO, resulting in the generation of nanometric fibers with diameters ranging from 50 to 250 nm [1,2]. Recent scientific studies have focused on the development of electrospun DNA nanofibers that can be further used in various processes and devices, for example for photonics and nanotechnology. Maleckis et al. [83] used a salmon-derived DNA of approximately 1,2 MDa molecular weight, dissolved in aqueous Tris and Ethylenediaminetetraacetic acid (EDTA) buffer. The electrospun nanofibers with a high molecular alignment, controlled diameters, and exceptionally good mechanical properties were obtained, indicating the potential application of DNA-based nanofibers as building blocks in various modern nanostructured devices. Szukalski et al. [84] synthesized nanofibrous optical switches based on a DNA bio-matrix and pyrazoline derivative. The DNA was functionalized with the cetyltrimethylammonium (CTMA) surfactant, doped with the luminescent and photoisomerizable chromophore, and was dissolved in a butanol solution. The resulting switches were to be evaluated for the design and development of fast and efficient sensing systems, photonic logic operators, and organic multiplexers.

### 7.3. Phospholipids

Phospholipids are molecules composed of a charged hydrophilic polar head and a hydrophobic tail. They function as the main building blocks of the cell membranes. Lecithin, a mixture of phospholipids and neutral lipids, can form cylindrical or wormlike micellar aggregates in nonaqueous solutions. At high concentrationd, the micelles entangle and can be successfully electrospun from the chloroform and N, N’-dimethylformamide (DMF) solution [1,2,40]. Asolectin is another candidate from the phospholipid family showing electrospinning potential. Derived in nature from soy, asolectin is built from lecithin, cephalin, and phosphatidylinositol building blocks, along with minor fraction of polar lipids and other phospholipids [85].

An unusual aspect of the phospholipid electrospinning process is that it is governed by the particles’ tendency to self-assemble. Hence, the typical empirical equation describing the correlation between the fibers’ diameters and the concentration entanglement and solution concentration can no longer be applied and the process is instead governed by the micelles tendency to agglomerate [21]. Asolectin microfibers, electrospun with limonene and isooctane as solvents, were proven to generate successful antioxidant matrixes that can be used to permanently encapsulate various phenolic compounds, such as vanillin and curcumin, for their delayed release under the aqueous conditions [86]. It can be also considered for use in various biomedical fields, where the local delivery of various active substances is desired. In addition to the previously described examples, a variety of polymers that contain phospholipid polar groups can be processed by the ES technique. The 2-methacryloyloxyethyl phosphorylcholine (MPC) polymers can be spun upon dissolution in chloroform and ethanol as solvents. The resulting material exhibits excellent antithrombogenic properties, making it particularly interesting in porous membrane applications in various blood filtration systems [56].

In 2015, Jørgensen et al. [21] investigated the impact of the processing parameters (phospholipid concentration, solvent type, spinning in standard, and co-axial ES) on the morphologies and diameters of the azolectin fibers. It was discovered that the fibers formed when the phospholipid concentration exceeded 45%. Specifically, fibers were obtained at 45 and 50% from DMF, 50% from cyclohexane and 60% from limonene and isooctane, with the latter showing the most uniform fiber diameter distribution. Furthermore, the fiber’s diameters were correlated with the solvent’s evaporation point and dielectric constant. It was also observed that at too high a concentration (above 65%), the ES cannot be conducted as the gelation occurs. By applying co-axial ES with an outer needle filled with DMF, a reduction in the fibers’ diameters was observed, approaching values calculated theoretically (without the micelle aggregation contribution). This was likely due to its high dielectric constant and DMF inducing the bending and Coulombic stretching of the jet, causing a thinning of the fibers and an improved uniformity. The authors hypothesized that the co-axial approach may be used to improve the spinnability of various biopolymers; however, this theory is yet to be confirmed.

In summary, phospholipids are emerging as a novel, promising, class of biopolymers. Unfortunately, they self-assemble and do not follow the typical empirical equations used in the ES experimental space calculations, making them challenging to the process. As a result, research focused on their electrospinnability is still rather rare.

### 7.4. Proteins

The next group of biopolymers frequently used in the ES is proteins, the backbone of all living organisms. Proteins often have a complicated, three-dimensional structure, created with strong intra- and intermolecular bonds. Consequently, their electrospinning poses many challenges. One of them is the identification of proper solvents and careful selection of the environmental parameters, not to induce an uncontrolled denaturation. As in most cases, the high effort required for the process of optimization pays off through the resulting remarkable structural and functional properties, such as the high natural extracellular matrix biomimicry and elasticity, making them optimal candidates to be used in medicine and in other related sciences [1,53,87]. Proteins are natural scaffolds of all living organisms and represent the main component of ECM fibers in all tissue types, hence their electrospinning is likely the most popular biomedical application. Protein-based polymers can be derived from animal tissues, plant extracts, or can be synthesized by living organisms as part of reconstruction and growth.

#### 7.4.1. Collagen and Gelatin

Collagen and gelatin are the most prominent examples of electrospinnable proteins. Collagen is one of the main components in connective tissue ECM. It is hardly soluble in water and as such, could prove to be challenging to ES. To overcome this potential roadblock, collagen is typically spun with the addition of PEO or is blended with polycaprolactone (PCL) to enable collagen’s native conformation retention. Chitosan—collagen blends are specifically mentioned, suggesting that the presence of polyanion–polycation interactions between the two compounds creates easily spinnable systems [88,89,90]. Alternatively, HFIP or 2,2,2-trifluoroethanol (TFE) can be used as solvents, yielding spinnable, optimally viscous solutions. However, due to the aggressive, denaturing nature of the previously mentioned solvents, the collagen’s native structure can be affected by the creation of the hydrogen bonds between the chains. As a consequence, water-soluble fibrous products are obtained that require the subsequent crosslinking. Less aggressive solvents are believed to maintain some of the native conformation, while still providing high solubility. An example of such an approach can be found in a study by Le Corre-Bordes et al. [30], where a methodic evaluation of the spinnability windows of citric (CA) and acetic acids (AA) used as solvents for collagen in the form of denatured whole chain collagen (DWCC) was performed. It was identified that AA was able to unfold the DWCC, while CA caused its folding. As a result, a broader ES range was found for the former. It was also discovered that DWCC maintained its native conformation, having a higher share of α chains than the gelatin [30]. In a follow-up study, CA was used as an efficient DWCC cross-linker for the process [52]. Kitsara et al. [49] tested three different forms of collagen: atelocollagen, acid fibrous collagen, and basic fibrous collagen, for their spinnability from the salt buffer solution and the ethanol (1:1) solvent system. CA was used as a crosslinker, while glycerol and sodium hypophosphite (SHP) served as an extending agent and catalyst, respectively. Only the atelocollagen was found to be spinnable, resulting in material characterized by an excellent cytocompatibility with cardiomyoblasts, allowing for good cellular adhesion and infiltration. In vivo studies on the mouse models revealed that the material can be easily handled and implanted into the heart, supporting regeneration of healthy tissue after the myocardial infarction. While long-term studies are still to be performed, the material is identified as particularly interesting, as the use of only biocompatible solvents and additives presents significant advantages over the other proposed scaffolds. In attempts to maintain a higher share of collagen’s native structure, Bazrafshan and Stylos [91] performed a grafting polymerization technique, using varying ratios of methyl methacrylate (MMA) and ethyl acrylate (EA). The modified collagen was then spun from the formic acid solution. While all of the tested MMA/EA grafting ratios were spinnable, a 1/3 ratio was able to maintain the collagen’s native conformation to the highest extent. A completely different approach was proposed by Wakuda et al. [44] In their study, collagen was dissolved in acetate buffer and electrospun inside the electrospinnable polyvinylpyrrolidone (PVP) polymer. The resulting material formed a core-shell composite. The collagen core was then gelated, and the PVP shell was washed off, giving rise to pure collagen fibers of a desired morphology and native structure. The tested material provided excellent results with human umbilical vein endothelial cells (HUVECs).

When the native structure of collagen is denatured through hydrolysis, gelatin, a biodegradable and biocompatible polymer, is obtained. While most of its properties are similar to collagen, it can be easily identified by decreased biomimetism, inferior mechanical properties, and higher water solubility, that unfortunately does not enable electrospinability from aqueous solutions due to its high surface energy and its high degree of hydrogen bonding. Therefore, solvents such as TFE or HFIP are employed to enable the stable ES process of defect-free materials [1,2]. To reduce the toxicity risk, other solvents combined with polymers were evaluated. For example, Yao et al. [92] electrospun gelatin from formic acid by adding the poly(vinyl alcohol) (PVA), enabling a stable ES process of the gelatin/keratin blend, yielding high-quality fibers. In this material, keratin was used to improve the material’s biocompatibility in vivo, resulting in an effective wound healing system. To enhance the mechanical properties of gelatin, a variety of polymers/biopolymer blends and additives were evaluated. Hivechi et al. [93] modified the solution with cellulose nanocrystals (CNCs). At a 5% concentration, the additives increased the material’s Young modulus and tensile strength over 3-fold, with no negative effects on the cytocompatibility. Even though the material seemed to have promising properties, its biological performance will have to be assessed more thoroughly. In another study by Jiang et al. [33], gelatin enhanced the cytocompatibility of PCL, yielding materials supportive for human mesenchymal stem cell adhesion, proliferation, and growth.

The amine-containing side groups of the collagen and gelatin can be modified with photo-cross-linkable methacrylate groups, leading to materials called ColMA and GelMA, respectively. While methacrylated collagen is widely tested for biomedical applications in the form of hydrogel [94], its usage for ES purposes is rather scarce. This is most likely because modification through methacrylation, followed by ES processing, leads to a structural denaturation, yielding GelMA instead of ColMA. Nevertheless, some articles claim the successful ES of ColMA. For example, Song et al. [95] had modified the collagen with methacrylic anhydride, dissolved the product in HFIP, and then performed the ES. The solution of the unmodified collagen served as a comparison. Both materials exhibited fibrous morphologies and were found to be water-soluble—the latter was inhibited by UV-crosslinking, only in the case of ColMA. The fact that the materials were soluble in water and no evidence of the peptide structural analysis was provided suggests that the final product was likely GelMA, rather than ColMA.

Methacrylated gelatin (Figure 5), on the other hand, is a very popular electrospinnable material, characterized by high biocompatibility and satisfactory morphological and chemical biomimetism to ECM fibers. Additionally, it promotes neovascularization, water adsorption and has a fast and easily controllable biodegradation rate. Thus, it has been widely used in tissue engineering applications in the form of a hydrogel scaffold [35,36,37]. Typically, GelMA is electrospun using HFIP as a solvent. Post ES, the material needs to be thoroughly dried under a vacuum to remove any solvent residues, and then soaked in photoinitiator solution, followed by a UV crosslinking. This procedure has two possible flaws: the risk of toxic solvent residues and the incomplete crosslinking of thick and dense materials when the photoinitiator is not able to fully penetrate the scaffold. Hence, alternative strategies are being designed. For example, Aldana et al. [54] dissolved the GelMA in acetic acid, supplemented with a I2959 photoinitiator. Hence, a less toxic solvent was used and a uniform distribution of the photoinitiator was achieved. This solution was then electrospun onto micropatterned molds used as collectors. Defect-free GelMA nanofibrous mats with highly biomimetic morphologies and controlled roughness were obtained. Thus, Aldana et al. proposed a rapid and inexpensive technique for sophisticated and complex scaffold formation, which might be found to be superior for biomedical applications.

#### 7.4.2. Elastin

Elastin is another example of fiber naturally occurring in the animals’ ECM. This insoluble, elastomeric protein is formed by crosslinking of its precursor, tropoelastin. Due to its high biocompatibility and interesting mechanical properties, fibrous scaffolds made of elastin are of high interest in wound healing, cardiovascular, and lung tissue regeneration applications. However, pure elastin is hardly spinnable and is most commonly spun with other polymers, from tropoelastin, or by using elastin-like recombinamers (ELRs). An extensive review of the latter can be found in a recent article by Cabello et al. [96] A study by González de Torre et al., on the other hand, is an excellent example of new achievements in this field [97]. The authors were able to synthesize ELRs that were in situ “clickable”, i.e., able to self-crosslink upon the ES, by forming chemical interactions between the two types of functional groups present at their chains: amine and azide. The materials, spun from the TFE solution, were stable in water and were found to promote the keratinocytes and fibroblast adhesion, proliferation, and growth. The orientation of the fibroblasts was found to align with the fibers, a property highly desired in the field of tissue engineering. Thus, the obtained materials seem to be very promising skin tissue engineering scaffolds.

Among the polymers blended with elastin, polyurethanes (PUR), PCL, and collagen can be listed, some of them being the subject of separate reviews [98,99]. In research by Heiny et al. [100], elastin was blended with PUR, obtained from polymerization of l-lactide and poly(ethylene glycol), all dissolved in HFIP. Post ES, the materials were crosslinked with GTA vapors, leading to fibrous, highly elastic, and biodegradable materials that could be applied as vascular grafts or wound dressings. Blends with up to 40% of elastin were found to be spinnable, and elastin was able to significantly enhance the tensile modulus of the PUR, while improving the adhesion and viability of HUVECs cultured on them. While further analysis on the material’s biocompatibility and stability is needed, the presented results are very promising.

In an article by Chong et al. [101], a collagen-elastin-PCL fibrous scaffold was obtained from the HFIP solution and was crosslinked in the GTA vapors. The addition of elastin increased the scaffold’s elasticity. In vitro results indicated that both human keratinocytes and human dermal fibroblasts adhered readily to the obtained materials, with good cellular infiltration and proliferation up to 28 days of culture. Subcutaneous implantation in the mouse model showed that the obtained materials performed better than the commercially available skin substitute, Integra^®^ —better cellular infiltration, and faster neovascularization, combined with mild inflammatory reaction were achieved. The authors concluded that this promising material should be subjected to further studies using animal models of higher skin tissue compatibility with humans, i.e., pigs.

Tropoelastin has been electrospun from the HFIP solution by Rnjak-Kovacina et al. [102,103], both separately and as a collagen-elastin blend. The obtained scaffolds were highly elastic, porous, and had good cytocompatibility. The materials with a high porosity were found to support the fibroblasts adhesion, infiltration and growth, and were suggested for the use in wound healing applications. In the in vivo experiments, the composite fibers were well-tolerated up to 6 weeks post-implantation in the subcutaneous implantation mouse model, supporting enhanced tissue regeneration, with the synthesis of collagen and capillary formation.

#### 7.4.3. Silk Fibroin

Silk fibroin (SF) is a fibrous protein extracted from the silkworm larvae cocoons, most typically, from *Bombyx mori* or *Antheraea assama.* Having a fibrous structure, SF should be prone to ES without additives and the feasibility of this process has already been proven in the literature, by performing ES from the formic acid [104,105,106] or aqueous solutions [107]. Still, as the process is challenging, the majority of the studies report performing electrospinning with various co-polymers: collagen [108], PEO [109,110,111,112], or PVA [113,114]. Post-electrospinning, the fibers can be stabilized by methanol/ethanol [104,105,106,108,109,110,111], EDC-NHS [107], or GTA + HCl [113]. The indicated stabilization mechanism of the alcohols or EDC-NHS is the transition of the amorphous structure into β-sheet conformation, while GTA + HCl induces chemical cross-linking. Interestingly, Zhou et al. [114] and Serodito et al. [112], in their articles from 2019, indicated that there is no need for additional stabilization steps. While Zhou did not provide proof of a prolonged stability of the fibers, Serodito performed 7-day-long in vitro studies, by directly seeding the human cells from the periodontal ligament (hPDLs) onto the scaffolds. Scanning electron microscopy (SEM) images of the experiments revealed a maintained fibrous morphology. Hence, the applied ultrasounds were suggested to induce the β-sheet partial transition before ES, affecting the solution’s viscosity and stabilizing the resulting fibers. In the in vitro studies, SF fibrous scaffolds have been proven to enhance the murine L929 fibroblasts adhesion, proliferation, and growth [106,109,113], enhance the human fibroblasts and keratinocyte proliferation while improving the keratinocytes’ differentiation [105]. They were also favorable for the hPDL cells adhesion and proliferation [112]. When the SF-based scaffolds were used in vivo for treating wounds in rats [108,110] or mice [111], excellent biocompatibility, combined with enhanced wound closure, was reported. The SF fibers have been reported to behave as suitable bioactive compound carriers, such as polyphenols [108], manuka honey [111], silver ions [115], or antimicrobial peptides [105]. All in all, SF seems to be a promising material for the fabrication of various tissue engineering materials, especially for the treatment of burnt or infected wounds.

#### 7.4.4. Whey Protein Isolate (WPI)

Whey protein isolate (WPI) is a substance obtained during the production of cheese. In that process, caseins contained in the milk are precipitated by acids or the renin enzyme, creating the by-product: whey, made of lactose, whey proteins minerals, and organic acids. WPI is obtained by further filtration of this solution. The production of 1 kg of cheese generates about 9 kg of whey, considered as a waste in the food industry. Unfortunately, its storage in large quantities can negatively impact the environment. Therefore, possibilities of further processing of whey and its use in various fields of science and industry are investigated. So far, the proven applications are for diet supplements, in the form of whey concentrate or WPI. In the worldwide research centers, WPI is studied for its potential in the manufacturing of safe-for-consumption bio-coatings and/or biofilms, designed to protect the food and improve its quality. Only a few studies have considered WPI as a potential biomaterial to be used as drug carriers, bio-coatings, membranes, scaffolds, and hydrogels for in vitro cell cultures and tissue engineering [38,47,116,117,118,119,120].

Pure WPI is usually obtained by the membrane filtration of whey. It is a substance rich in globular proteins, especially β-lactoglobulins, α-lactoglobulins, immunoglobulins, and serum albumins (Table 2). Depending on the research needs, individual proteins can be further isolated from WPI by several methods, such as membrane filtration, selective adsorption, selective precipitation, and selective elution, all employing the physicochemical differences in the properties of molecules [117].

WPI exhibits extraordinary properties that make it particularly interesting for tissue engineering applications. First, it has favorable physicochemical parameters, such as optimal viscosity and high-water solubility. Second, it can emulsify and easily form three-dimensional hydrogels or foams, without requiring the use of chemical cross-linkers (which increases the purity and biocompatibility of the material). Specifically, a thermally induced cross-linking takes place, due to the formation of disulfide bridges between cysteine units and hydrophobic bonds between β-lactoglobulins [121]. As a result, the cross-linking can be combined with the sterilization procedure, increasing the cost-efficiency of the process. These properties are particularly interesting in the production of various types of scaffolds, e.g., for cell culture. WPI is also favored for being widely available, inexpensive, and biodegradable [38,47,116,117,118,119,120].

As it is for all low M_W_ and high-water soluble biopolymers, ES of pure WPI is also extremely difficult, or even impossible. In addition, the globular proteins are unstable, due to the lack of entanglement between the individual molecules of their chains. Therefore, for the ES of WPI, well-spinnable additives are utilized to act as the carrier for proteins. Only few compounds playing this role are reported in the literature. Among them, the most popular ones are PEO and two polysaccharides—maltodextrin (MD) and dextran (DE) [117,120].

PEO is a water-soluble thermoplastic polymer, commonly used as an additive in the electrospinning of various biopolymers. It has a satisfactory biodegradability and a low toxicity, contributing to its high popularity as an additive for the ES of food and biomedical industries products [42,117]. PEO enables electrospinning of WPI through several mechanisms, which are not yet fully understood. The first one is its ability to counteract the arbitrary charge formation in the ES solutions, known to hinder the ES of polyelectrolytes. The second is PEO’s ability to entangle its chain with the proteins, consequently increasing the solution’s viscosity. In the course of numerous experiments, it was observed that the spinnability of PEO/biopolymer solutions strongly depends on several factors, such as solvent type, biopolymer morphology, and the presence of substances that are capable of protein denaturation [39,117].

The research literature considers various WPI to PEO ratios, which allow for the stable and reproducible production of high quality, ultrafine fibrous products. Typically, by increasing the PEO content, the solution becomes more viscous and spinnable and the fibers obtained from mixtures with higher PEO ratios are usually thinner than those obtained from solutions with a higher WPI content. The pH value of the solution seems to be extremely significant as well, and should be kept as low as possible, as any increase leads to higher amounts of defects, including beads and discontinuous fibers, as presented in Figure 6 [39,42,47,117].

Despite its numerous advantages, PEO has not been fully approved for safe consumption, nor is it fully biocompatible. Therefore, other materials, with a similar effect on the ES process, are being evaluated for the WPI electrospinning [122]. Among these, polysaccharides of high M_W_ are often suggested. Their advantage over other additives lies in the fact that they are to increase the stability of the WPI by glycation of proteins via the first stage of the Maillard reaction. In this process, carbonyl groups of polysaccharides react with free ε-amino groups of lysine residues, which are the main glycation sites for proteins. As a consequence, covalent bonds between the sugar and the protein are formed (Figure 7). Glycoconjugates produced by the glycation of WPI and MD/DE show a high thermal stability, an increased ability to emulsify, and higher stability in the environments of different pH values, as compared to pure WPI [34,122,123].

In addition, the high M_W_ of MD/DE ensures the stable formation of the polymer fibers in an electrostatic field. Still, the optimization of M_W_ is challenging, as both too high and too low molecular weight of the polymer has been reported to prevent the successful ES. The literature indicates that M_W_ of 70 kDa is optimal for the MD or DE ES [34]. The selection of the polysaccharide source must be appropriate as well. In the case of MD, only the product that is obtained from potato starch by enzymatic hydrolysis was found to be suitable for electrospinning purposes, while the material derived from corn was not [34,122,123]. DE is biosynthesized from saccharose by the specific strains of bacteria, mainly *Leuconostoc bacteroides* and *Streptococcus mutans* [125]. Its molecules are chains of D-glucopyranose groups, connected to each other by α-1,6 glycosidic bonds, with many branches connected mainly by α-1,4 glycosidic bonds (Figure 8) [34,126]. As a result, DE is a highly flexible polymer, with high solubility in both water and organic solvents, and relatively low solution viscosity (compared to other biopolymers). DE does not form electrostatic complexes in contact with proteins, being a common phenomenon found in other polysaccharides [34,127,128]. Therefore, in our opinion, it is the most promising material to be used as an additive in the ES of WPI.

Currently, the WPI—MD/DE composites are not manufactured on the industrial scale due to the lack of economically efficient methods. The research is focused on synthesizing composite fabric materials via electrospinning. As presented by the literature evaluation, the production is currently carried out mainly by the needleless electrospinning technique, as presented on the scheme in Figure 1. While the process allows the generation of a significant amount of product in a short time and eliminates the clogging of the needle, it does not allow us to obtain the desired alignment and controlled fiber diameters. Hence, it is important to develop a method for the WPI—MD/DE composite manufacturing, that includes a needle-based electrospinning technique [34,122,123].

### 7.5. Other

As recent studies indicate, numerous types of proteins are now incorporated as blends with various polymers, to ensure desired properties of a resulting material. Among some of the popular, cost-effective, and green additives, keratin [129], soy protein isolate [130,131], zein [132], or casein [133] can be listed. However, to the best of our knowledge, neither of these materials has been reported to be independently electrospinnable. Instead, keratin was spun from the PVA blend, while soy protein isolate and casein were spun from either PEO or PEG modified solutions, with the final concentration of the protein in the fibers reaching up to 80%. It is worth noting that while all of the studies presented proof of fibrous morphology of the obtained scaffolds, most did not present the results of the materials’ long-term stability in the water environment. The exceptions in this matter are papers by Selvaraj et al. [133] and Esparza et al. [129], who provided SEM images of the cells growing on the surface of casein or keratin fibers, respectively, after up to 14 days of culture. In conclusion, for now, these materials present only hypothetical tissue engineering applications.

## 8. Paving the Way towards Clinical Applications—The Most Promising Fields of Application

Overall, due to their extraordinary physicochemical and biological properties, electrospun biopolymers are expected to be used in numerous applications in various fields of biomedicine, the most promising being pharmacology, tissue engineering, and regenerative medicine [5,9]. These are graphically depicted in Figure 9 below. Aside from biomedical applications, the food industry has been actively researching fibrous biopolymers for use in food processing and food packaging [6,57].

The main advantage of electrospun biopolymer fibrous matrices is their high resemblance to natural ECM, in terms of both morphology and chemistry. It has been already stated that such materials can provide an excellent environment for effective tissue growth and regeneration, as well as guiding the desirable cellular response. Excellent biomimetism and bioactivity are being harvested, mainly for use in tissue engineering, as three-dimensional scaffolds for tissue growth and substrates for cell cultures, as well as in regenerative medicine for wound healing purposes. Listed fields use proper architecture and specific chemism of polysaccharides, proteins and phospholipids to achieve a desirable cellular reaction, and promotes neovascularization and neo-tissue genesis. Furthermore, an electrospinning technique allows the user to control the porosity of the final product. Properly chosen porosity affects not only the cellular penetration of the biomaterial, but also establishes effective gas exchange and liquid absorption, which in the field of wound healing can be used to avoid desiccation, dehydration, and to absorb excess of wound exudate [8]. A high surface area-to-volume ratio of fibers allows for specific chemical or biological functionalization of the surface to further enhance the biomimicry of the material. Furthermore, this property, along with good biodegradability of most biopolymers, ensures the efficient encapsulation and release of various drugs and/or other functional compounds, widely used in pharmaceutical and wound healing applications [69]. Lastly, a key advantage of some biopolymers over other biomaterials lies in their native antimicrobial properties. Due to the increasing problem of drug resistance, observed in various microorganisms, intrinsic antibacterial properties are currently highly desirable [19].

Although electrospun biopolymer matrices are characterized by numerous properties that are extremely beneficial in various fields of biomedical engineering, they still possess numerous drawbacks. Due to their inferior mechanical performance, fibrous biopolymers are unhandy and cannot be considered for use in applications that require transferring significant amount of a mechanical load [8], with the exception of cellulose [80]. Moreover, the limited possibility to control the overall biodegradability in a biological environment further decreases the potential use in scaffolding, drug delivery, and wound healing applications. However, the biggest drawback is probably the fact that the biopolymer’s properties lack reproducibility from batch to batch [123], even when the same polymer from the same production series is evaluated. This may lead to a low reproducibility of the final product, and, as a result, create a significant drawback for major scientific and industrial companies. In conclusion, the process of biopolymer commercialization in medical-related fields requires a very careful optimization and a step-wise scale up.

To summarize, among various biopolymers, polysaccharides, e.g., cyclodextrins, chitosan, cellulose or starch, are to be considered, alongside with phospholipids, mainly for use in wound healing and drug delivery fields. They are fairly abundant, biodegradable, and their processing is based on green chemistry techniques, ensuring a high biocompatibility of the resulting material and allowing for safe encapsulation of various drugs and bioactive compounds without the risk of contact contamination. At the same time, some polysaccharides, e.g., kefiran [76], are characterized with bacteriostatic properties, allowing for safe use in environments that are highly susceptible to bacterial infection (mainly wounds). Proteins, on the other hand, seem to be particularly interesting in the fields of tissue engineering (as potential scaffolds) and cell cultures (as highly specialized substrates). Their high bioactivity and the fact that native ECM consist mainly of proteins like collagen [90] and elastin [96] makes them perfect candidates for applications where contact with cells and tissues are a key factor. It is worth noting, however, that a majority of biopolymers are not being processed on their own, but with the addition of other substances and mainly other polymers (both synthetic and natural [2]). Such an approach not only allows for the electrospinning of compounds that are hard or unspinnable on their own, but also allows the gain of composite materials with extraordinary and highly desirable properties, thus enhancing the versatility of each component in various biomedical fields.

## 9. Summary 

In recent years we have witnessed significant progress in the field of biopolymer ES. New materials are now successfully electrospun, with or without copolymer addition. Several new, eco-friendly additives and solvents are investigated, rendering the process green and more cost-efficient. The rising trend is to reduce the use of toxic organic compounds (such as HFIP) and in this aspect, salt buffer solutions or biopolymer additives such as copolymers are particularly interesting. From the biomedical point of view, much attention has been given to maintaining the materials’ native conformations, thus keeping the extremely high biocompatibility and biomimetism that allows the biopolymers to excel in various applications. Significant attention has also been given to the use of new crosslinking agents that would maintain the fibrous morphology while guaranteeing a low toxicity and sufficient stability (both in terms of mechanical properties and degradability).

When it comes to biomedical applications, the main areas of interest are wound dressings and tissue engineering scaffolds—the fields that benefit the most from the high biocompatibility, ECM biomimetism, and fibrous morphology. As the main fibrous compounds of the natural ECM, proteins are certainly the most interesting substrates to be used in medical applications. Among these, the most abundant ones: collagen (and its derivatives: gelatin or GelMA) and elastin (and its derivatives: ELRs) are of particular interest. They exhibit excellent cytocompatibility and the initial evaluation indicates their in vivo biocompatibility, as presented on the examples in Figure 10.

Additionally, silk fibroin also appears to be popular. Its main advantage over elastin and collagen is its lower price, due to higher abundance and much simpler processing. However, the use of SF requires caution as research indicates its tendency to induce an enhanced inflammatory response, depending on the processing and composition of the product [136]. New trends in tissue engineering also call for the use of materials’ mixtures that would combine all the major constituents of the native tissue, enhancing the cellular differentiation and maturation.

Considering the economic and ecological points of view, the use of by-products of various industries, such as the food industry, is particularly tempting. It is our opinion that the use of this class of materials will become of increasing significance. In particular, biomedicine can benefit from the biocompatible wastes created in the food fabrication processes. One example of such a material is WPI, which has already paved its way as a dietary supplement but now increasingly proceeds into the creation of a new class of tissue engineering scaffolds. Certainly, further progress in processing it into desired shapes (i.e., fibers) is required, but the future looks bright in this matter.

While ES is a fairly straightforward and versatile technique, able to produce fibrous morphologies out of various biomaterials, it is important to always keep in mind its various limitations. First, it does not allow the production of sterile products. This is particularly troublesome in the case of biopolymers, where only limited sterilization techniques exist that do not cause their degradation. Most often, one is forced to resort to ionizing radiation methods, which are expensive and not as widely available as autoclave. Troublesome sterilization may significantly increase the production costs of medical products. Secondly, the repeatability of the production process may be limited in the case of biopolymers which may vary from batch to batch—stringent quality control is needed. Next, the ES process leads to a large charge accumulation at the material itself. This might be hard to discharge, cumbersome to handle and can alter the biological performance of the products. Finally, while there are some strategies to yield 3D constructs [18,137], this is still a very challenging task that yields fragile products. As a result, ES is limited to obtaining fibrous meshes. On the other hand, very thin layers of fibers are challenging to remove from the collector without physical damage. In summary, even though the use of ES to obtain fibrous, biopolymer-based products is tempting due to its versatility and steerability, its application in the manufacturing of the product of desired features and quality should always be given careful consideration. Not all biopolymers will prove to be electrospinnable and for those that will, a strict and thorough optimization would be necessary to obtain fibrous, defect-free, and properly crosslinked materials of a desired conformation. With a properly developed process, materials of superior quality and a versatile applicability could be obtained.

The strength of this review lies in its thorough critical analysis of the literature regarding current advances in the electrospinning of biopolymers to be used as biomaterials. Even though the subject is extremely broad, in some cases, the available literature is scarce and not all the biopolymers are given the same attention, with the largest emphasis being placed on proteins. In our opinion, this is the biggest drawback of the published research pool. Furthermore, some research summaries present contradictory results, and no universal strategy is presented to guarantee a high quality, electrospinnable biopolymer manufacturing approach. As this is an emerging field, there is more work to be done to guarantee the successful fabrication of high-quality electrospun, biopolymer-based biomaterials that in the future may enter the path to clinical trials, followed by clinical applications.

## Figures and Tables

**Figure 1 jfb-12-00026-f001:**
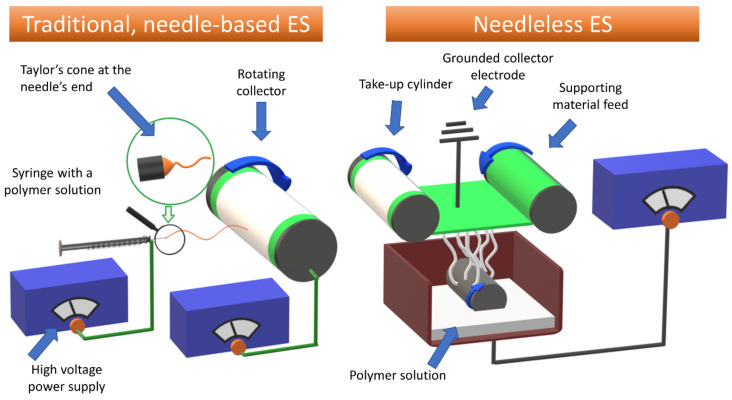
Two of the most popular electrospinning (ES) setups: needle-based (**left**) and needleless (**right**). Setup with rotating mandrils allowing for a high throughput and versatility is depicted in both cases.

**Figure 2 jfb-12-00026-f002:**
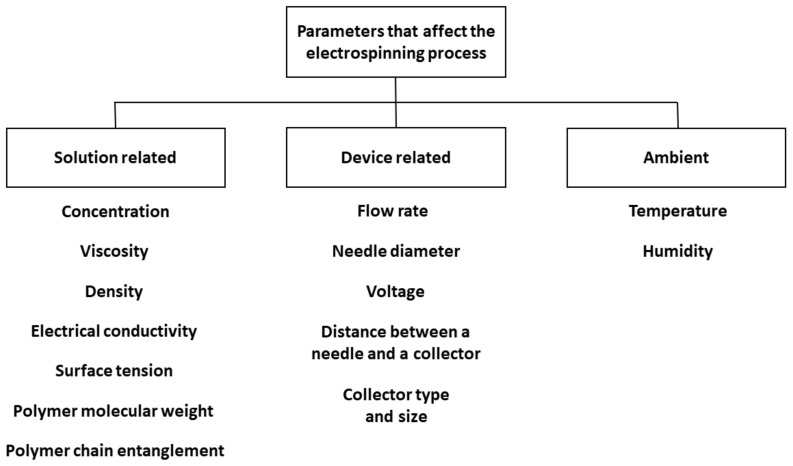
The parameters affecting the ES process, segregated by the category.

**Figure 3 jfb-12-00026-f003:**
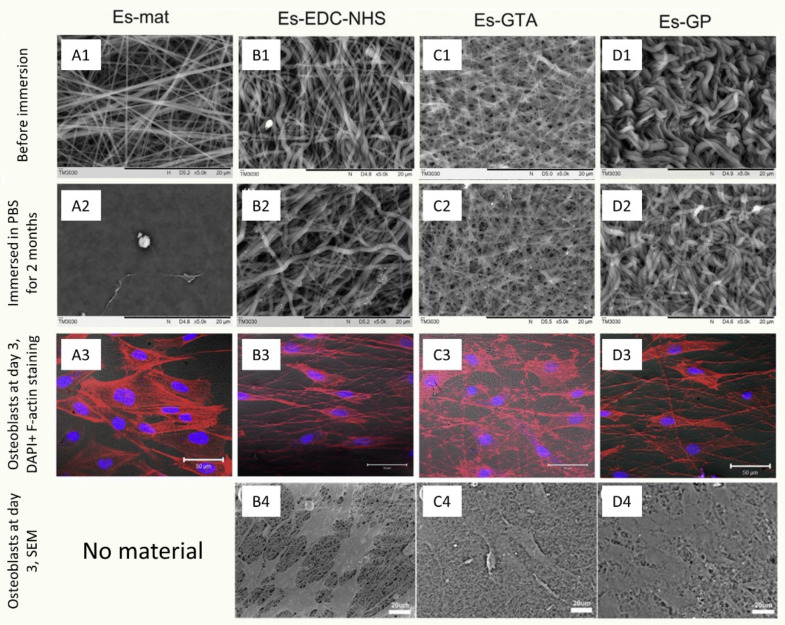
Impact of various types of chemical crosslinkers on the morphology (**A1**–**D1**) and chemical stability (**A2**–**D2**), biocompatibility (**A3**–**D3**), and 3-day stability (**A4**–**D4**) of the ES collagen nanofibers. Reprinted from [50] with permission from Elsevier, copyright 2018.

**Figure 4 jfb-12-00026-f004:**
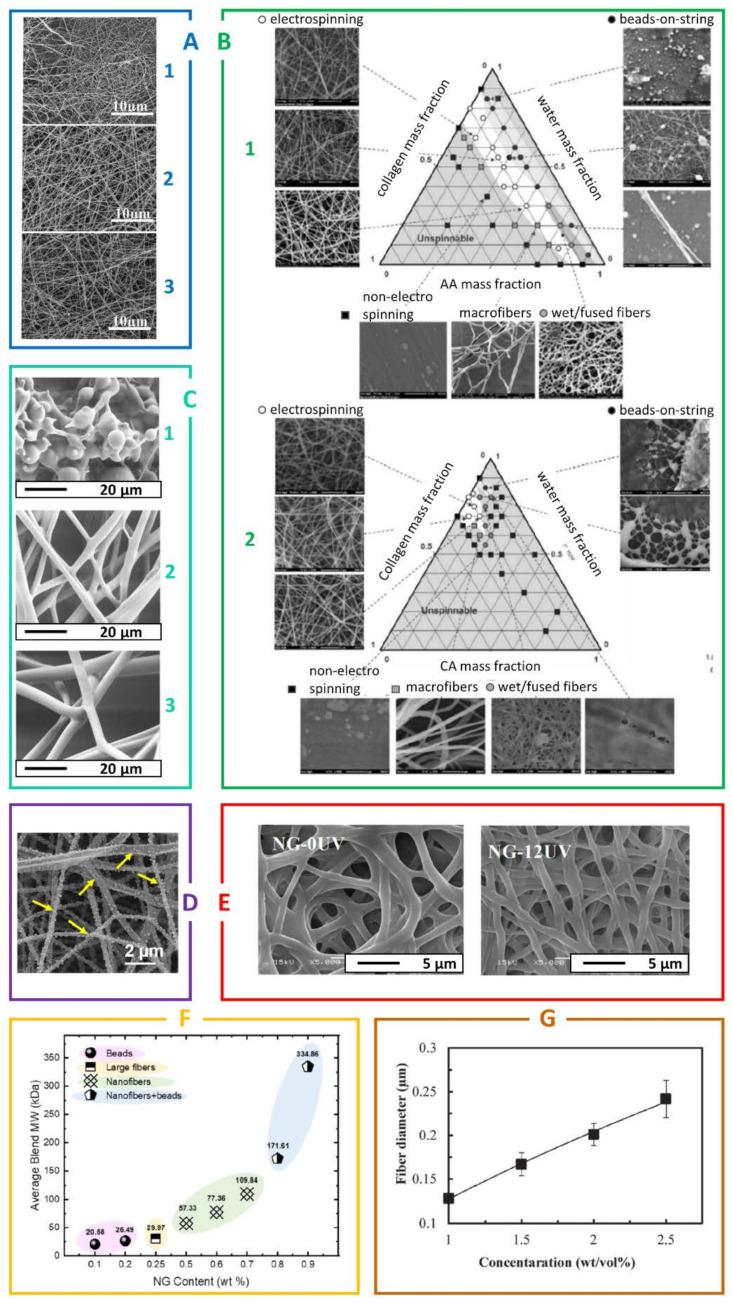
Examples of the optimization studies geared towards identification of the biopolymers’ electrospinnable windows and adequate post-processing. (**A**) SEM images indicating the relationship between the fraction of the xanthan polymer during processing: (1) 1.5 wt/vol%, (2) 2.0 wt/vol% and (3) 2.5 wt/vol% and the resulting fibrous morphology. Reprinted from [32] with permission from Wiley, copyright 2017. (**B**) Tertiary diagrams illustrating the correlation between the solvent fraction (acetic and citric acid respectively) in a denatured whole chain collagen (DWCC) polymer system and the morphology of the electrospun products. The diagrams aid in the identification of the optimal solution composition yielding defect-free fiber products. Reprinted from [30] with permission from Elsevier, copyright 2018. (**C**) The FESEM images presenting the relationship between the concentration of the phospholipids in the initial solution: (1) 35 wt%, (2) 45 wt%, (3) 50 wt% and the fibrous morphology from the resulting polymer product. The significant change in the morphology, from micellar to cylindrical and fibrous, is observed. Reprinted from [40] with permission from Science, copyright 2006. (**D**) An example of the material electrospun from the salt solution: SEM image of pullulan fibers (15% PUL; 1.0 M NaCl) with crystals of salt visible on the surface. Reprinted from [31] with permission from MDPI, copyright 2017. (**E**) SEM images of the fibrous morphology stabilized by the introduction of the light-initiated crosslinking—electrospun GelMA fibers prior to UV light (NG-0UV) and after 12 min of UV exposure (NG-12UV). Reprinted from [54] with permission from MDPI, copyright 2019. (**F**) Diagram illustrating the correlation between the optimal biopolymer concentration and molecular weight and formation of the fibrous products. The optimal guar content with moderate molecular weight ensures fabrication of bead-less fibers; both too high and too low values of MW and concentration yield a mixture of thick fibers and beads. Reprinted from [41] with permission from ACS Omega, copyright 2019. (**G**) Linear function of the fiber diameter as a function of the biopolymer (xanthan) concentration in the spinning solution. Reprinted from [32] with permission from Wiley, copyright 2017.

**Figure 5 jfb-12-00026-f005:**
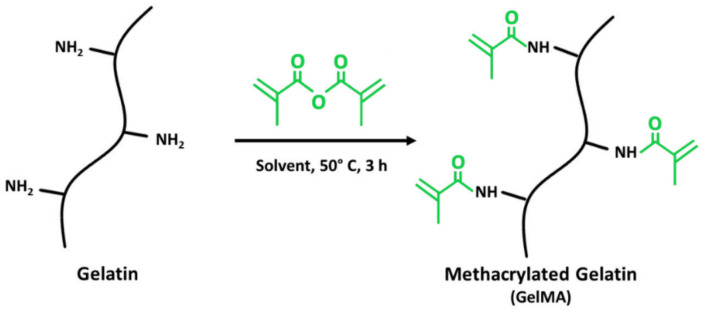
A schematic representation of the GelMA fabrication process.

**Figure 6 jfb-12-00026-f006:**
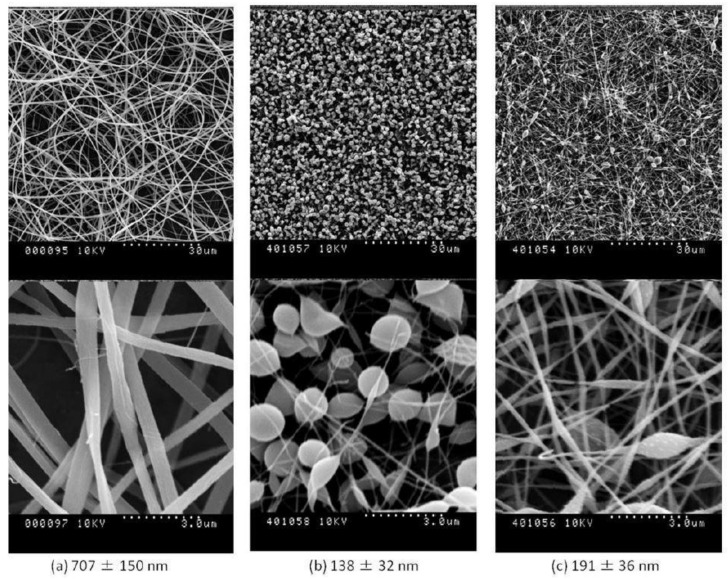
SEM images of WPI-PEO electrospun nanofibers obtained from solutions at: (**a**) pH 1; (**b**) pH 7; (**c**) pH 12, observed under the increasing magnification. Reprinted from [39] with permission from Wiley, copyright 2012.

**Figure 7 jfb-12-00026-f007:**
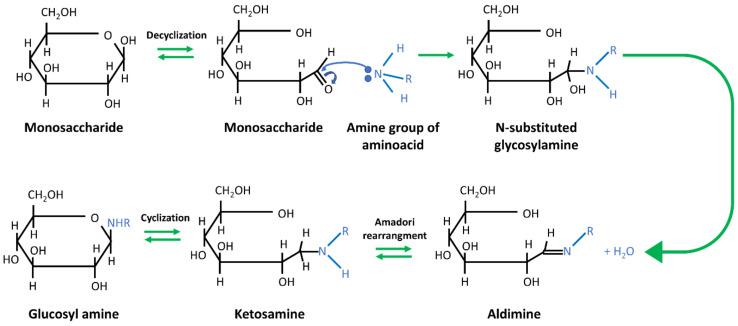
Scheme of the initial phases of Maillard reaction—the D-glucose molecule. Based on [124].

**Figure 8 jfb-12-00026-f008:**
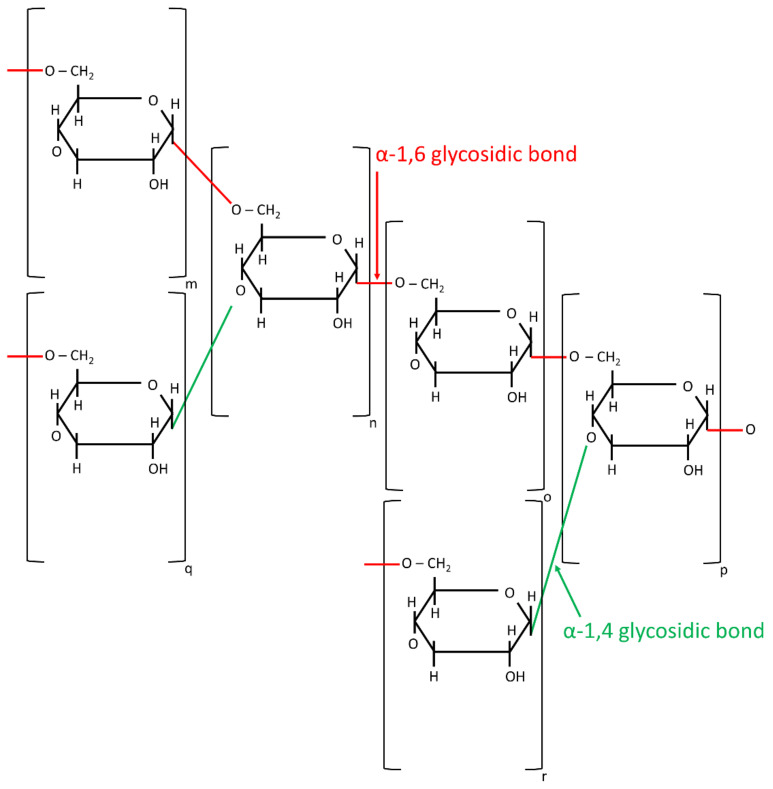
The scheme of a dextran polymer chain.

**Figure 9 jfb-12-00026-f009:**
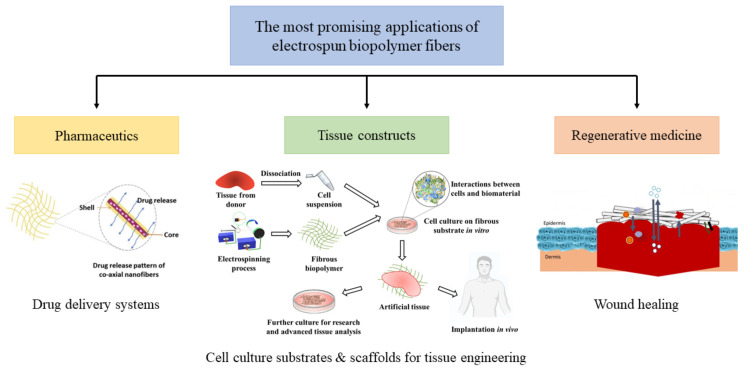
The illustration of the most promising applications of electrospun fibrous biopolymer matrices: (**Left**) Pharmaceutic field as drug delivery system. Drug release pattern of co-axial nanofibers with active compound encapsulated inside the core [10]. (**Center**) The use of biopolymer fibers as cell culture substrates and scaffolds for tissue engineering (fibrous biopolymer—[10]; culture dish—[134]; interaction between cells and biomaterial—[135]; human posture and Eppendorf—Free sources). (**Right**) Wound healing application in the field of regenerative medicine. The application of fibrous biopolymer membrane on skin wound and interactions relevant for wound healing that might occur due to porosity of the material, i.e., gas exchange and liquid absorption [8]. Reprinted from [8] with permission from Elsevier, copyright 2020.

**Figure 10 jfb-12-00026-f010:**
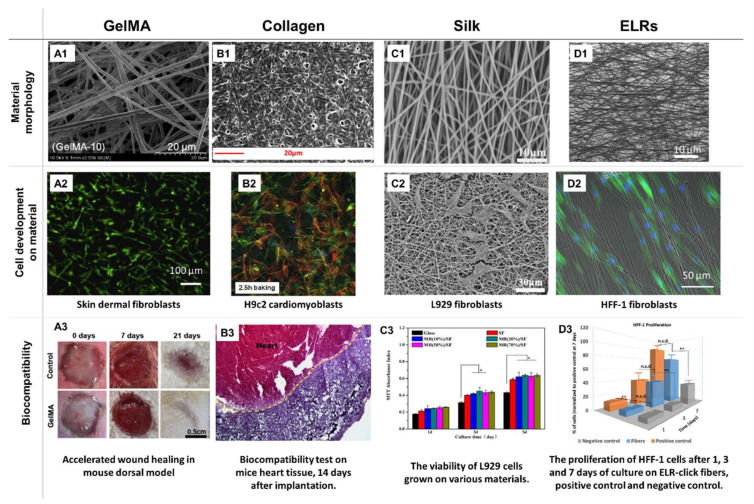
The examples of the excellent biocompatibility of the electrospun biopolymeric scaffolds. Electrospun GelMA nanofibers, cross-linked by 10-min UV exposure (**A1**) and skin dermal fibroblasts on day 7 after seeding on material’s surface (**A2**). In vivo results revealing an accelerated wound closure efficiency induced by the material (**A3**). Reprinted from [35] with permission from Elsevier, copyright 2017. Image of electrospun collagen fibers after cross-linking in water with 10% citric acid (**B1**). Confocal images of H9c2 cardiomyoblasts on the electrospun collagen scaffold after 48 h of cell culture (**B2**). Image of material biocompatibility towards mice heart tissue 14 days after implantation (**B3**). Reprinted from [49] with permission from Elsevier, copyright 2015. SEM image of silk nanofibrous matrix with 10% addition (*wt*/*v*) of manuka honey, tread with 75% ethanol vapor (*v*/*v*) (**C1**). Image of L929 fibroblast cells on the very same silk nanofibers (also with 10% (*wt*/*v*) manuka honey) (**C2**). The viability of L929 cells grown on different substances used in the research (**C3**). Reprinted from [111] with permission from Elsevier, copyright 2017. Morphology of electrospun ELR-click fibers (deposition time = 90 s), cross-linked by thermal treatment in water (**D1**). Phalloidin and DAPI staining of oriented HFF-1 cells on ELR-click fibers (**D2**). Proliferation histograms of fibroblasts (HFF-1) after 1, 3, and 7 days of culture on ELR-click fibers (FIBERS), positive control, and negative control (**D3**). Reprinted from [97] with permission from Elsevier, copyright 2018.

**Table 1 jfb-12-00026-t001:** Some of the strategies used to remediate the electrospinnability problems in various biopolymer solutions.

Biopolymer	Strategy	Mechanism	Citation
Xanthan	Formic acid as a solvent	Esterification reaction between the formic acid and pyruvic acid groups of xanthan	[32]
Lecithin	Raising the phospholipid concentration	As the lecithin concentration rises, the micellar morphology changes, and the cylindrical micelles overlap and entangle	[40]
Guar	High M_w_ native guar electrospun with its partially hydrolyzed low M_w_ analog	Synergistic effect leading to stable nanofiber formation	[41]
WPI	PEO (polyethylene oxide) as additive	PEO as the polymer with high electrospinnability properties serves as a carrier for peptide molecules	[42]
WPI	Dextran as additive	Maillard reaction resulting in obtaining WPI-dextran-conjugates	[34]
Starch	Aqueous/DMSO solution as a solvent	Dimethyl sulfoxide (DMSO) establishes a sufficient polymer chain entanglement	[43]
Pullulan (PUL)	Salt’s aqueous solutions as solvents	Salt metal ions disrupted the hydrogen bonds in PUL and altered its solution properties, increasing the viscosity and polymer chain entanglement	[31]
Collagen	Polyvinylpyrrolidone (PVP) as a spinnable shell	ES of core-shell fibers, with spinnable material used as a shell, removed after the ES process is completed	[44]

**Table 2 jfb-12-00026-t002:** Physical properties of main proteins found in whey protein isolate, based on [117].

Protein	Molecular Mass (kg/mol)	Concentration (g/L)	Isoelectric Point (pI)
β-lactoglobulins	18	3.2	5.4
α-lactoglobulins	14	1.2	4.4
Immunoglobulin G	150	0.7	5.0–8.0
Serum albumin	66	0.4	5.1

## Data Availability

Because of the character of the study (review), no data could be shared at this time. The authors do not hold the rights to copyright the data from the presented research articles.

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
