# Peer review of "Advances in Fabricating the Electrospun Biopolymer-Based Biomaterials"

_jfb, 2021, doi:10.3390/jfb12020026_

Round 1

Reviewer 1 Report

In the descriptions of the electrospinable biopolymers the attention is mostly paid to proteins . It would be interesting to know more about phospholipids and polynucleotides.
Descriptions of some figures are not clear need corrections.

Author Response

Indeed, proteins are in a focal point of this review and this is not without a reason. This class of biomolecules constitutes main components of the mammalian tissue’s ECM. As such, their usage in fabricating novel biomaterials is the most popular and also the most promising area of research. Particularly, when one’s goal is to create biomimetic scaffolds, using the same components that are found in the native tissue seems to be the method of choice. At the same time, when the material is appropriately processed into the desired morphology and shape that allows maintaining its biofunctional nature, the cells can recognize it as a natural constituent of their environment, which in turn can lead to improved biological performance - enhanced proliferation, differentiation and/or proliferation. 

Hence, not only are the studies concerning biomedical usage of proteins more popular than it is in the case of any other biopolymers, but also are their results far more promising. What is more, this group of materials also has a very large share of electrospinnable representatives. Both of these factors have led to visible imbalance of our study, with the largest attention being paid to proteins. We have added a brief justification of this into the introduction. Still, we’ve also tried to add some more examples of other biopolymers so that the peptides are slightly better weighted out. 

The figure captions have now been corrected

Reviewer 2 Report

Recommendation: Major revisions

Comments: 

This manuscript describes the recent advances in techniques used for making various biopolymers and and biomedical applicability of the electrospinnable biopolymers. The authors need to address the following comments and revise the manuscript accordingly. 

  1. Please consider to include and highlight different techniques including TEM, AFM, DLS for the characterization of the assembled structures. Please consider to include the following references.
    • Roy, D.; Park, J. W. Journal of Materials Chemistry B, 2015, 3 (26), 5135-5149. “Spatially nanoscale-controlled functional surfaces toward efficient bioactive platforms”.
    • Xue, J., Wu, T., Dai, Y., & Xia, Y. (2019). Electrospinning and Electrospun Nanofibers: Methods, Materials, and Applications. Chemical reviews, 119(8), 5298–5415. https://doi.org/10.1021/acs.chemrev.8b00593
  1. Consider to highlight functional self-assembled biopolymers and their applications. 
  2. Please consider to add macromolecules like dendrimers/dendrons etc. and include the following references. 
    • Paez, J.I.; Martinelli, M.; Brunetti, V.; Strumia, M.C. Dendronization: A Useful Synthetic Strategy to Prepare Multifunctional Materials. Polymers 2012, 4, 355-395. https://doi.org/10.3390/polym4010355
    • Roy, D.; Kwak, J.-W.; Maeng, W. J. et al. Langmuir, 2008, 24, 14296-14305. “Dendron-Modified Polystyrene Microtiter Plate: Surface Characterization with Picoforce AFM and Influence of Spacing between Immobilized Amyloid Beta Proteins”.
  1. Please consider to highlight specific biomedical applications and include a table.
  2. Please highlight biopolymer conformation in dispersions and their processing properties including fiber forming mechanisms on a molecular level.

Author Response

  1. Please consider to include and highlight different techniques including TEM, AFM, DLS for the characterization of the assembled structures. Please consider to include the following references.

    Even though we must agree with the reviewer that identifying proper evaluation techniques of the electrospun materials is needed, we must argue that an extensive guidance of these was not an objective of this particular study. We feel that such analysis would unproportionally increase the article’s volume, while not introducing too much necessary information to the subject it concerns. Hence, instead of listing the available techniques in this study, we’ve now added a guidance where such information can be found. This information is now introduced as chapter 6. We do hope that this solution would satisfy the reviewer.  

  2. Consider to highlight functional self-assembled biopolymers and their applications.

    This has been corrected by introducing a new paragraph: Page 5, line 205

  3. Please consider to add macromolecules like dendrimers/dendrons etc. and include the following references.

    While we agree with the reviewer that these are interesting types of materials, we must argue that modifying the polymers with dendrimers or dendrons is hardly an ES strategy. Instead, it is sometimes used to grant the materials with new functionalities post-spinning and this has now been added into the study, giving cellulose as an example: Page 15, line 579

  4. Please consider to highlight specific biomedical applications and include a table.

    This has now been added as a section 8 of review. Instead of a table, a graph was created.

  5. Please highlight biopolymer conformation in dispersions and their processing properties including fiber forming mechanisms on a molecular level.

    For the sake of the article’s volume, in the first version of the article we did not analyze these problems in full as they weren’t the main subject of the study. However, as we recognize that the study will likely reach a broad audience, some of which might not be familiar with the subject, we’ve now added a short section at the end of chapter 3, as per reviewer’s instructions highlighting the need of achieving a certain biopolymer’s conformations in the solution which guarantee sufficient molecular entanglement. However, we must stress that only basic information is given and references where further information can be found are added.  We do hope this would satisfy the reviewer as we really would not want for our study to deviate from the subject too much.

Reviewer 3 Report

Review of manuscript: “Advances in fabricating the electrospun biopolymer - based biomaterials”

Journal of Functional Biomaterials (jfb-1129434)

This paper deals with an interesting topic, which may potentially contribute to filling some gaps in the existing literature in the area of biopolymers and biomedicine. The authors did a very good job in  summarizing many related publications on this topic and they provided essential information about the current status of the electrospun biopolymer - based biomaterials. However, I have a number of minor concerns regarding the review discussion and presentation, which are as follows.

Minor comments:

  1. The introduction should conclude with a clear subsection not only showing the key messages and aims of this paper, but also emphasising the significance of the contribution of this review. Also, please clarify what it adds to the subject area compared with other already published reviews.
  2. It would be beneficial if the authors could discuss the features, applications and limitations of electrospun biopolymer - based biomaterials and their role in biomedical applications in a short separate section.  A chart, which describes biomedical applications of electrospun biopolymers, would be also helpful to add.
  3. In the summary section, it should be highlighted what are the strengths and the weaknesses of this review as well as its limitations. Please write also a short paragraph with clear conclusions consistent with the presented evidence and arguments. They should address not only main advances in the electrospun biopolymer - based biomaterials, but also the limitations of electrospinning. 
  4. The paper is generally well writen and there are only some very small style issues (for example, om page 4, in line 152, replace the phrase "What is more, ..." with "Furthermore, ..." or "Moreover, ..."

Author Response

  1. The introduction should conclude with a clear subsection not only showing the key messages and aims of this paper, but also emphasising the significance of the contribution of this review. Also, please clarify what it adds to the subject area compared with other already published reviews.

    We would like to thank the reviewer for this valuable comment. The introduction section has now been changed by adding two new paragraphs. The first one reviews the information presented in various reviews that concern the subject of choice, identifying areas which are not yet sufficiently analyzed. The second paragraph emphasizes the idea behind this review and justifies the need for it.

  2. It would be beneficial if the authors could discuss the features, applications and limitations of electrospun biopolymer - based biomaterials and their role in biomedical applications in a short separate section. A chart, which describes biomedical applications of electrospun biopolymers, would be also helpful to add.

    This has now been added as a section 8 of review. Instead of a table, a graph was created.

  3. In the summary section, it should be highlighted what are the strengths and the weaknesses of this review as well as its limitations. Please write also a short paragraph with clear conclusions consistent with the presented evidence and arguments. They should address not only main advances in the electrospun biopolymer - based biomaterials, but also the limitations of electrospinning.

    The authors would like to thank the reviewer for his/hers valuable remarks which improves the quality of the article’s summary. Based on that remark, two new paragraphs were added to the summary section.

  4. The paper is generally well writen and there are only some very small style issues (for example, om page 4, in line 152, replace the phrase "What is more, ..." with "Furthermore, ..." or "Moreover, ..."

    Thank you for this positive opinion. This issue has now been addressed.

Round 2

Reviewer 2 Report

The manuscript is revised. However, moderate editing and English improvement are required. Publish after minor revision.